# Agent-Based Modeling for Integrating Human Behavior into the Food–Energy–Water Nexus

**Nicholas R. Magliocca**

Department of Geography, University of Alabama, 513 University Blvd., Box 870322, Tuscaloosa, AL 35401, USA; nrmagliocca@ua.edu

**Abstract:** The nexus of food, energy, and water systems (FEWS) has become a salient research topic, as well as a pressing societal and policy challenge. Computational modeling is a key tool in addressing these challenges, and FEWS modeling as a subfield is now established. However, social dimensions of FEWS nexus issues, such as individual or social learning, technology adoption decisions, and adaptive behaviors, remain relatively underdeveloped in FEWS modeling and research. Agent-based models (ABMs) have received increasing usage recently in efforts to better represent and integrate human behavior into FEWS research. A systematic review identified 29 articles in which at least two food, energy, or water sectors were explicitly considered with an ABM and/or ABM-coupled modeling approach. Agent decision-making and behavior ranged from reactive to active, motivated by primarily economic objectives to multi-criteria in nature, and implemented with individual-based to highly aggregated entities. However, a significant proportion of models did not contain agent interactions, or did not base agent decision-making on existing behavioral theories. Model design choices imposed by data limitations, structural requirements for coupling with other simulation models, or spatial and/or temporal scales of application resulted in agent representations lacking explicit decision-making processes or social interactions. In contrast, several methodological innovations were also noted, which were catalyzed by the challenges associated with developing multi-scale, cross-sector models. Several avenues for future research with ABMs in FEWS research are suggested based on these findings. The reviewed ABM applications represent progress, yet many opportunities for more behaviorally rich agent-based modeling in the FEWS context remain.

**Keywords:** sustainability; simulation modeling; large-scale behavioral modeling; integrated water resources management; nexus-plus

## 1. Introduction

Interdependencies among food, energy, and water resources pose contemporary and future sustainability challenges, and have become research and policy priorities conceptualized as the nexus of food–energy–water systems (FEWS) [1–4]. FEWS nexus thinking focuses on identifying and analyzing interconnections between systems (e.g., coupled in their supply, processing, distribution and use) with the aim to support nexus-wide rather than single sector decision-making [2,5]. Managing change at the FEWS nexus, often with the intent to transition to more sustainable FEW resource use patterns, is a multi-scale coordination challenge. Growing populations, shifting climate patterns, and new technologies are constantly pressuring current resource supply and demand relationships, and require coordinated interventions across FEW sectors and between system- and actor-levels within each sector [1,4,5]. However, FEWS research efforts have predominately met the challenges of multi-sector analysis with a macroscopic perspective by aggregating or abstracting spatial and actor heterogeneity within each sector, and consequently producing knowledge that is difficult to implement in any specific place or at the level where distributed production or consumption actually occurs [1,6]. Alternatively,

more finely resolved research efforts addressing spatially explicit and/or disaggregated production or consumption behaviors are often unable to scale-up insights to entire resource systems, which inhibits integration with and analysis of feedbacks to/from other sectors [6–8].

Computational modeling has been the dominant tool for formalizing and quantifying FEWS nexus relationships due in part for their potential to explore implications of policy or technology interventions before their widespread implementation [9–11]. Recent advances in FEWS modeling improve our ability to conceptualize, analyze, and manage multi-scale, FEWS interconnections, but many limitations persist (see reviews by [7,12,13]). In particular, there has been limited inclusion of human behavior and social interactions in FEWS modeling that are known to influence supply/demand relationships, production/consumption decisions, and pro-environmental behavior [10,14–16]. For example, environmental psychology theories of individual behavioral change consider a suite of factors, such as attitudes, norms, perceived behavioral control, awareness, and a sense of personal responsibility to explain decisions to adopt environmentally friendly technologies [17–20]. Social networks have also been demonstrated to be a powerful influence on individual technology adoption decisions in agricultural contexts [21]. While empirical data on the drivers of individual behavioral change is essential, linking such information to changes in regional resource use footprints requires innovative aggregation methods. Efforts to represent regional-level behavioral change in FEWS modeling have relied on assumptions of a representative consumer (a group), perfectly informed choices based on rational optimization, and instantly equilibrating markets [22]. Omitting individual-level behavioral factors, which can act as barriers or motivations for technology adoption, from FEWS analysis risks misleading estimates of regional- or sector-level resource use efficiency and trade-offs.

This gap in current FEWS modeling illustrates three key challenges as described by McCarl et al. [4]. First, choosing the appropriate level of complexity at which to model each FEW sub-system must balance adequate representation of sub-system component heterogeneity (e.g., farmers) with compatibility to other FEW sub-system models. The second, related challenge is integrating multiple sub-models that often have different spatial and/or temporal resolutions. For example, farmer decision-making around water use may be represented at the farm-level, whereas hydrologic models may span regional river basins. Finally, models that are most useful for decision-making support and stakeholder engagement are those that can accommodate a diversity of alternative policy options, while also representing the stakeholders being modeled with sufficient realism to gain buy-in during participatory modeling exercises [23,24].

Agent-based modeling (ABM), also referred to as multi-agent systems (MAS) in engineering disciplines, is a tool that can address all three of these challenges. ABM addresses the first two challenges through an inherent multi-level structure–system-level outcomes emerge from interactions among heterogeneous and adaptive individuals [22,25]. Thus, diversity in individual behavioral factors, decision-making, and social and environmental contexts can be modeled explicitly, which produce aggregate regional resource footprints that are compatible inputs for larger-scale FEW sub-system models. ABM can address the final challenge by representing realistic differences among individual attributes and behavioral factors, rationally bounded decision-making, and the influence of social processes on technology adoption decisions, leading to simulations that "look right" to stakeholders [24]. Ultimately, this allows ABM to explore the effects of alternative scenarios at the regional level as the result of interactions among individual entities, and inform the design of policy incentives to overcome diverse behavioral barriers to transition to production/consumption patterns that maximize benefits across FEW systems.

Building on needs identified by recent reviews of FEWS modeling and related ABM approaches, this review considers both the research contexts and simulation designs of ABM applications in FEWS research. This review represents a horizon-scanning effort to consolidation information about the design, implementation, and motivations for using ABMs in FEWS research, and does not address the broader implications of specific FEWS interactions, policies, and/or scenarios. The research context pertains to broader publication trends of ABM applications, motivations for using ABMs, FEWS sectors

and intra- and inter-sector interactions that were modeled, and the use of empirical data to inform the models. The simulation design relates to modeling practices and choices about what is modeled, how it is modeled, and integration of social science insights into the model representation. ABM applications were reviewed to address the following research questions:

Research context:

- Which sectors and types of cross-sector interactions are modeled?
- For what purposes are ABMs used, and what are the justifications for using ABMs over other modeling approaches?
- To what extent do quantitative and qualitative data inform model development?

Simulation design:

- What spatial and temporal scales, for both simulated environments and agents, are represented?
- Which agents are modeled and what drives their behavior?
- How are human behaviors and social interactions incorporated into and influence cross-sector interactions?

In the process of addressing these questions, directions for future research are suggested to enrich the application of ABMs in FEWS research.

## 2. Materials and Methods

A literature search was conducted using Web of Science, Science Direct, and Google Scholar with keywords applied to the entire articles. The search was constrained to articles written in English and published in peer-reviewed journals, but no restrictions on publication date were imposed. The steps taken in the search process are presented in Figure 1.

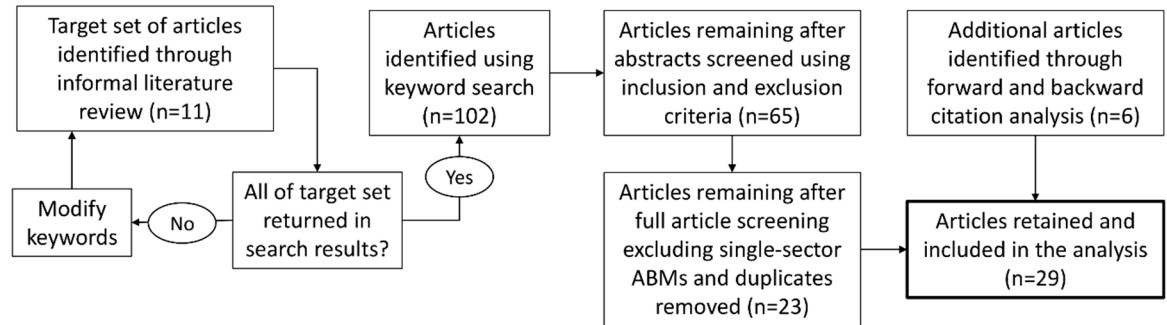

**Figure 1.** Search and screening process to assemble the article collection for review.

A "target set" of 11 known articles was assembled to represent the full expected range of ABM publications in the FEWS research domain. The target set was assembled with articles in journals from a wide range of disciplines and with both true and false positives to maximize the breadth of the literature search. False positives included articles with single-sector ABMs (e.g., water resources management [26]), processes related to FEWS, but not explicitly analyzing FEWS interactions (e.g., irrigation technology adoption [27]), and conceptual model frameworks without implementing an actual model [28]. Target set articles are listed in Appendix A. Keywords were assembled from the target set articles to initiate the formal literature search. The final list of search terms used was as follows:

"('agent-based' OR 'multi-agent') AND model* AND (((food OR agri*) AND water) OR ((food OR agri*) AND (biofuel OR *energy)) OR (water AND (biofuel OR *energy))) AND (system* OR nexus) AND (supply OR demand OR market OR efficien* OR sustainab*)"

Search terms included both "agent-based" and "multi-agent" variations of the modeling approach, possible FEWS combinations, and concepts related to FEWS nexus analysis, such as market supply

and demand, efficiency, and sustainability. The list of keywords was modified and the search repeated until all articles in the target set were contained in the search results (Figure 1). The resulting dataset consisted of 102 publications.

The abstracts of the publications returned in the search results were next screened using basic exclusion criteria. Articles were excluded if they did not (1) include ABM as at least one modeling approach used, and (2) present an implementation of a model or model formalization. The latter criteria was used to exclude publications that presented only a conceptual modeling framework or approach (e.g., [16,28]). While such publications were helpful for conceptualizing ABM challenges in the broader FEWS context, they were not (nor intended to be) formalized to the degree that a model could be reproduced from the description provided. For example, a publication was excluded if it did not define agents' specific attributes or decision-making rules. Importantly, standardized model documentation (e.g., overview, design concepts, and details + decision-making (ODD+D); [29]) and/or evaluation was not required for inclusion, because many model implementations did not provide more than a narrative description. Sixty-five articles remained after this initial screening.

A second round of screening was performed using the full text of each remaining article. Articles were primarily screened to determine if at least two sectors were explicitly considered in the ABM and/or overall modeling approach. The article must have addressed the research questions related to interactions and/or trade-offs between two or more food, energy, or water sectors. For example, many ABM applications focused on water resource systems (see [30]), and analyzed environmental impacts, infrastructure design, and/or policy in response to shifting water supply and/or demand without explicitly analyzing the effects of those shifts on food or energy production/consumption. Consistent with usage in the wider FEWS research domain [1,2], food systems were broadly conceived as agricultural systems, ranging from subsistence to industrial farming and staple to bioenergy crops. Fisheries were also considering in this definition of food systems, but no ABM applications in that domain were still included at this stage. Excluding articles with single-sector ABM applications reduced the data set to 23 articles. Finally, we conducted backward and forward citation analysis with Google Scholar. Six more articles were included after applying the inclusion and exclusion criteria, which increased the total number of articles to 29 used in the review (Supplementary Table S1).

## 3. Results

### 3.1. Research Context

The number of ABM applications in FEWS research has been increasing over the last decade, and nearly half of the reviewed articles were published in the last three years (Appendix B, Figure A1). Roughly 20% of articles were published in 2020. The earliest publication was in 2008 [31], which predated the Bonn Nexus Conference 2011 that approximates the point in time when the FEW nexus became a mainstream research and policy topic [2]. The 29 ABM applications reviewed were published in 24 outlets (Table A1). Many of the outlets were sustainability-focused and/or specialized in single sectors, although several interdisciplinary journals and a few modeling journals were also present. The relatively small number of modeling journals present suggested that ABM is viewed as a viable and innovative tool in the FEWS research domain, rather than the development of multi-sector, FEWS models as an advance in the ABM approach.

The inherent flexibility of the ABM approach was evident in the diversity of FEWS connections that were modeled (Figure 2). ABM applications were categorized by the specific cross-sector connections they modeled, and which sector was prioritized as an entry point. Typically, one sector was modeled in greater detail than another, and/or scenarios or trajectories of change among two or more sectors were analyzed as cascading changes in one sector causing and receiving feedback from another sector or sectors. Overall, food–energy and food–energy–other connections were the most frequent application of ABMs (12) (Table 1). The energy sector was the primary entry point for "food–energy" studies, which prioritized the analysis of bioenergy crops (both first- and second-generation crops were

considered) and their potential expansion as the catalyst for cross-sector changes. Potential trade-offs with food production were considered in the forms of competition for land, displacement of food crop production leading to subsequent land-use change, and/or food security impacts. Net greenhouse gas (GHG) emissions [32,33] and environmental impacts, such as soil erosion [34] and loss of wildlife habitat [35], were additional sectors considered (Figure 2).

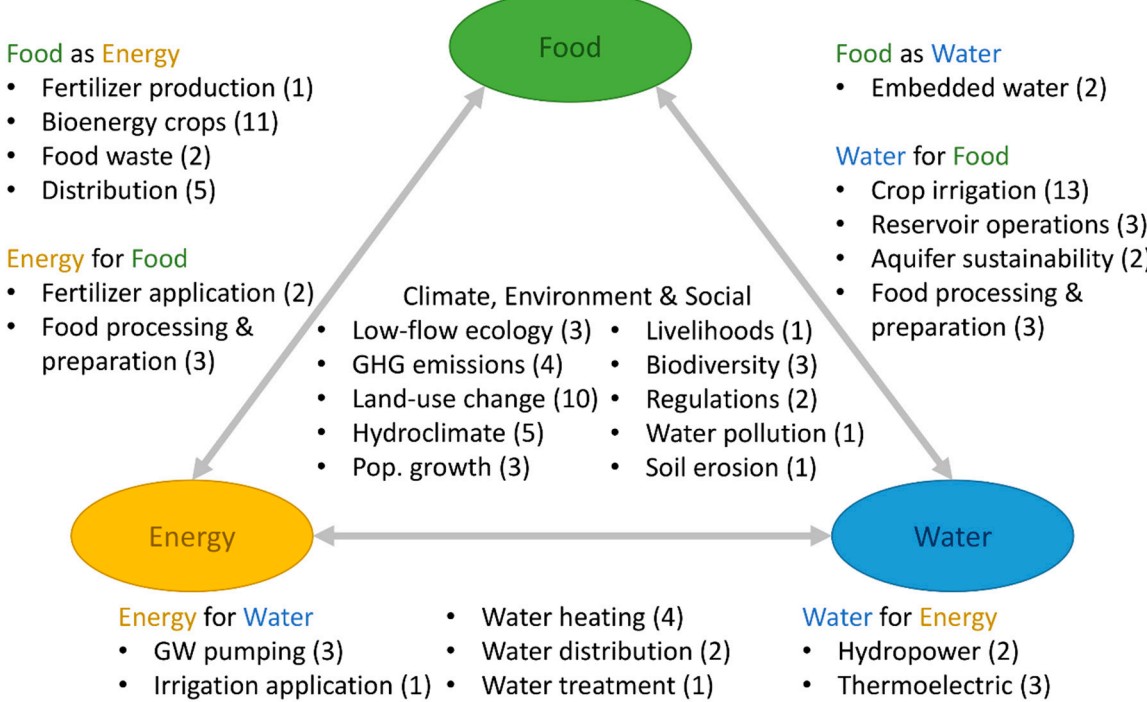

**Figure 2.** Food–energy–water system (FEWS) linkages studied in the reviewed agent-based modeling (ABM) applications. Additional interactions with climate, environmental, and/or social impacts were also present.

**Table 1.** Cross-sector connections explicitly modeling in ABM applications in FEWS research. The "FEWS Priority" column indicates which sector was the primary entry point.

| FEWS Sectors | Articles | FEWS Priority |
|---|---|---|
| Food–Energy | 7 | 0-7 |
| Food–Energy–Other | 5 | 2-3-0 |
| Energy–Water | 2 | 0-2 |
| Energy–Water–Other | 1 | 1-0-0 |
| Food–Water | 5 | 3-2 |
| Food–Water–Other | 4 | 3-1-0 |
| Food–Energy–Water | 1 | 0-0-1 |
| Food–Energy–Water-Other | 4 | 2-0-2-0 |
| **Total** | **29** | **29** |

The second most frequent application of ABMs was for food–water sectors (9) (Table 1). All applications considered agricultural production activities in some form, but the simulation scales varied considerably ranging from individual farmers to cities (see Section 3.2.2). Crop selection as a driver of or in response to irrigation adoption animated many of these studies. Other recurrent themes included impacts to stream hydrology [36] and water use conflicts [31] in the context of shifting water availability under future climate (i.e., hydroclimate) and/or population growth (Figure 2). Energy–water ABM applications (3) were all household-level simulations of interrelated water and energy demands generated by daily in-home and out-of-home activities [37–39]. The ABM applications

that addressed additional sectors or dimensions beyond FEWS were generally equally split between environment (7) and greenhouse gas emissions (7) and one focusing on resource-related conflicts [31] (Table 1). Environmental dimensions included vegetative cover changes and impacts on hydrology [36], water, and/or air quality or soil erosion from agricultural practices [32,34,40,41], maintaining minimum river flows for ecological processes [8], or modifications to wildlife habitat [35].

Finally, five ABM applications considered all three FEW sectors, and four of those considered additional problem domains. The larger percentage of models in this latter category reflects the structural modeling designs that were required to consider interconnections among FEW sector. Once the infrastructure was built to incorporate all three FEW sectors, an additional dimension could be achieved with marginally less model design changes.

### 3.1.1. Justifications for Using ABM

The most frequently invoked justification for using an ABM approach was the recognition that human behavior is an important driver of FEWS dynamics and a current limitation other FEWS modeling efforts. However, human behavior was taken to mean different things across applications. Generally, all applications interpreted interactions among agents as critical to model and understand how real FEWS change, and ABMs were seen as the most suitable modeling approach for representing and analyzing interactions [34,41,42]. Several articles emphasized the importance of individual motivations and coordinated interactions among independent stakeholders for creating, and thus modeling, emergent collective behavior [8,39,41,43]. Other behavior-based justifications included the need to represent non-economic motivations for recreating market dynamics, such as boom and bust cycles in bioenergy markets [42,44], which are not captured with assumptions of rational behavior and/or market equilibrium in other FEWS modeling approaches [43]. More generally, there was a recognition that ABMs are best suited to represent human behavior that varies across individual attributes and/or environmental heterogeneity, because statistical distributions rather than just the averages can be used in large cross-sector or transboundary systems [8,35,45,46].

An equally frequent justification for the use of ABMs was their distributed architecture, which enabled realistic representation of spatially distributed resource supply and demand [37]. Spatial heterogeneity and the effects of distributed decision-making were often cited as important explanations for non-uniform opportunity costs or constraints on production choices, resource demands, or adoption of resource-efficient technologies [35,36,46]. Nikolic et al. [47], Khan et al. [48], and Yang et al. [8] cited difficulties in accounting for spatial heterogeneity in systems dynamics models as a limitation in integrated water resources management (IWRM) contexts. A distributed architecture was also needed to better account for the interactions among agents when there was directionality to interactions, such as upstream and downstream water users [31], or producers/consumers in material flow networks between farms, and with their upstream and downstream partners [32]. Accordingly, the ease with which spatial heterogeneity could be represented and integrated with GIS in ABMs was seen as a strength of the approach [38].

The complexity of FEWS was frequently acknowledged and ABMs cited as the best option to handle such complexity. Theoretical frameworks and research methods for complex systems, such as Complex Adaptive Systems (CAS) theory and ABMs, respectively, were invoked several times as a basis for solving problems in dynamic, interconnected systems with multiple objectives and high uncertainty [49]. FEWS were explicitly labelled as CASs several times in the context of supply chains, which were described as complex networks of producers and processors/consumers in food systems [32] and analyses of bioenergy economic viability/competitiveness [42,50]. ABMs offered the potential to generate interaction structures, such as supply chain networks, from individual interactions to simulate sector-level outcomes [40]. This was seen as an advantage of ABMs allowing the simulation of alternative dynamic pathways through bottom-up interactions, which cannot be captured by other top-down modeling approaches [42].

The inherent multi-level structure of ABMs, consisting of system-level behaviors that emerge from individual interactions, was seen as particularly amenable to multi-scale FEWS problems.

The agent-based approach was explicitly chosen because of the "intrinsically modular" design, which facilitated the aggregation of distributed supply and demand to regional or sector scales [51] (p. 4). Dermody et al. [52] and Agusdinata et al. [42] both chose the agent-based approach because of the cross-scale implications of distributed decision-making in interconnected FEWS, which can be individually rational and optimal, but can create sub-optimal outcomes at the system scale and/or in another sector. Allain et al. [6] also cited limitations in current modeling assessments of water management strategies, which do not include spatial and temporal scales large enough for decision-making and sustainability issues, while simultaneously considering finer resolution interactions among agents.

Finally, precedents for using ABM to conduct scenario analysis and recommend corrective policy action [53,54] were cited as an advantage of ABMs. Zhuge et al. [38] suggested that the distributed architecture of ABMs made it more useful for policy assessment and infrastructure planning, because simulations can be constructed at high spatiotemporal resolutions. Falconer et al. [55] and Nouri et al. [56] asserted that policy recommendations generated by top-down modeling methods may not be effective, because they fail to account for agents' behavioral norms and potential changes in behavior in responses to interventions. As pointed out by Schulze et al. [57] and Bieber et al. [10], many FEWS policy questions focus on introducing or expanding a production activity for which few or no studies exist, and the agent-based approach enables exploration of these activities by simulating decision-making processes and their implications under different or hypothetical conditions. Similarly, Holtz and Pahl-Wostl [58] emphasized the process-based nature of ABMs as an advantage, which enables causal connects between changing conditions and behavioral responses and/or system transitions.

### 3.1.2. Model Purpose

Closely aligned with the justifications for using ABMs were the stated model purposes. Model purpose is the first element of the overview, design concepts, and details + decision-making (ODD+D) protocol, which is current praxis for documenting ABMs [29,59], and a crucial guide for model design choices [60]. The purpose for developing a model will have implications for how realistic agents and their interactions need to be, and potential trade-offs between general applicability versus case study fidelity. The model purpose also sets the objectives against which the modeling exercise is judged. The stated purposes of FEWS ABMs ranged from relatively basic research along the lines of increasing system understanding and methodological innovations, to more applied purposes of scenario analysis, prediction, and policy recommendations. Table 2 provides an accounting of stated modeling purposes and their associated articles.

**Table 2.** Total count and associated articles that stated a given modeling purpose. Many articles stated more than one purpose, in which case an article will appear in more than one category.

| Purpose | Count | Articles |
|---------|-------|----------|
| Methodological Innovation | 18 | [6,8,31,35–41,43,45,47,48,51,52,55,56] |
| System Understanding | 17 | [6,10,32,33,35,36,38,42–45,48,50,52,55,57,58] |
| Scenario Assessment | 12 | [6,32,37,40,42,46,49–51,56,57] |
| Inform Policy | 9 | [33,34,37,45,46,51,55–57] |
| Prediction | 3 | [34,39,49] |

To some degree, all of these ABM applications could be considered methodological innovations, since ABMs are not yet mainstream in FEWS research [28,55]. Therefore, "methodological innovation" was only recorded when it was explicitly stated as a model purpose. Even with that restriction, authors recognized the methodological innovations required to integrate ABM into FEWS research, and thus "methodological innovation" was the most frequently stated model purpose. Methodological innovations were most often in coupling ABMs with other modeling approaches to simulate interactions between two or more sectors and/or scale-up distributed activities to sector or regional outcomes. Coupling with

hydrological models was common, and the soil and water assessment tool (SWAT; [61]) was the most common hydrological model used for its ease of translating agent changes in land use to SWAT inputs [6,8,41,48]. Nouri et al. (2019) simulated groundwater withdrawals for irrigation with MODFLOW [62], and other surface water models included the HEC-HMS modeling system [63], CatchScape3 [64], and two unspecified and/or custom hydrological models [36,52].

Outside of the water sector, a similar mix of "off-the-shelf" models and custom formulations were used to meet specific sector modeling needs. Two ABM applications were integrated with individual-based models to simulate impacts of agricultural land-use changes on forest structure [36] and skylark populations [35]. Allain et al. [6] used the AqYield model to simulate crop growth and yields resulting from farmers' crop choices. Partial equilibrium models for either the agricultural sector or as part of larger integrated assessment models were used in applications that explicitly considered market prices [46,52]. Several ABM applications integrated optimization models implemented at the agent-level for profit maximization [34] or investment allocation decisions [10,43], or at the system-level to optimize flows [39,40] or facility locations [55] in supply networks. Finally, Zhuge et al. [38] integrated MATSim and SelfSim microsimulation models to generate high-resolution water and energy demands from agents' daily activities.

System understanding was also a commonly stated model purpose. The modular and distributed architecture of ABMs frequently cited as justifications for their use were aligned with the desire to investigate interconnections between FEW systems at the level at which supply and demand originate–individual consumers and producers. A key knowledge gap that ABMs were often used to address was the scaling-up of individual-level and spatially distributed resource use/supply activities to provide dynamic and spatially explicit forcing within and across sectors. This cross-scale link was the primary advance in system understanding cited as resulting from ABM applications relative to other modeling approaches. A second related aspect of system understanding enabled by ABMs was the study of FEW system-level dynamics that resulted from collective behavior and coordination among individual actors. Such social behaviors were often cited as a gap in other modeling approaches and important processes in driving sector-level consumption or production patterns [34,37,39,41,43].

Scenario assessment was cited as an explicit modeling purpose in just under half of the articles. ABMs were seen as well-suited for scenario analysis, particularly in contexts in which few or no contemporary examples exist [10,57], because agents' decision-making could be represented at the process-level rather than prescribed based on rules derived from past observations of decision outcomes (e.g., regression analysis). Consequently, agents could respond to changing conditions (e.g., bioenergy crop subsidies) according to a consistent goals (e.g., profit maximization) and set of decision rules that accommodated a variety of in- and out-of-sample conditions. Scenarios most frequently assessed the potential outcomes of alternative policy interventions. The effects of bioenergy crop subsidies, taxes, or other price changes were recurrent scenarios explored in energy–food applications. Fine resolution ABM application (i.e., individual consumers, time steps of seconds to minutes) in energy–water applications assessed changes in individual-level consumption patterns resulting from the introduction of higher efficiency technologies and resource conservation measures [10,38]. Alternative climate scenarios were also commonly used to explore the impacts of altered hydroclimate on water availability and/or agricultural production.

Sensitivity analysis using the one-factor-at-a-time (OFAT) approach [65–67] was used in several applications in a scenario-like fashion to explore the impacts of uncertain and/or key parameters. For example, Fernandez-Mena et al. [32] analyzed the effects of varying social interaction parameters, such as agents' disposition for exchange and preferences for certain materials, on material flows in an agro-food network. Agusdinata et al. [42] used OFAT to demonstrate the sensitivity of biofuel supply chains to time delays in price information received by individual agents. Moncada et al. [50] tested the effects of varying the rate at which farmers adapted their future prices expectations based on past observations, which led to less adaptive price expectations and larger impacts on biofuel production from exogenous shocks. Tourigny and Filion [39] examined the effects of both agent- and market-level

parameters on adoption rates of low-flow fixtures, and found that agent-level adoption probability had the greatest impact on technology diffusion rates. Schulze et al. [57] took a slightly different approach to sensitivity analysis and explored the effects of alternative configurations of land productivity and processing plant locations throughout stylized landscapes. They found that under certain economic conditions, poorly sited processing plants could discourage short rotation forestry production.

Often accompanying scenario assessment were objectives of informing policy and/or predicting future production or consumption patterns. Indeed, the ability to inform policy by accounting for behavioral changes was one of the main justifications for using ABMs in FEWS research. Importantly, applications that were used to inform policy or predict possible outcomes acknowledged the uncertainty associated with constructing multi-scale and multi-sector models; however, little discussion was devoted to the implications of using models for these purposes, particularly given the politically-charged trade-offs often involved in FEWS [68,69].

### 3.1.3. Use of Empirical Data

As applications of ABMs have broadly moved from theoretical, "exploratory" models to multi-scale, multi-actor, and data-intensive models, empirical data of various kinds and collected with a range of methods is used to inform ABMs [70]. Empirically informing, calibrating, and validating agents' decision-making are persistent challenges for using ABMs, in FEWS research and socio-environmental systems (SES) modeling more broadly [71]. These challenges are particularly salient in the FEWS context when ABMs are designed to simulate the actions and decisions of many different types of agents across different sectors. It was unsurprising then to find nearly 40% of articles cited the lack of individual-level and/or behavioral data as a main ABM limitation.

No articles relied on a single data source to calibrate or inform ABM parameters, which speaks to the multi-faceted modeling challenge of FEWS research. Production (e.g., crop production, water) and consumption (e.g., electricity and water use) activities where the most frequent target for calibration with empirical data. Parameter values most often came from the relevant literature, and typically took the form of compiling specific parameters from multiple, diverse publications. Nearly as often "official" data sources from global (e.g., Intergovernmental Panel on Climate Change (IPCC) reports, Food and Agriculture Organization (FAO), World Bank), national, or regional statistical services were tapped for aggregate statistics about population demographics, housing characteristics, agricultural production, price, water resources, and climate and biophysical conditions. Geospatial data was often used and several articles explicitly mentioned integration with GIS platforms [36,44,55].

Finer resolution data for modeling agents and their interactions with one another or their environment were used sparingly. Parcel data consisting of georeferenced parcel boundaries linked to ownership information was used in only two applications. Allain et al. [6] simulated farmers' irrigation and crop management decisions within parcel boundaries associated with a single farm, which supported high fidelity, daily water withdrawal estimates across a roughly 800 km$^2$ watershed in southwestern France. Guillem et al. [35] used agricultural parcels in a 132 km$^2$ Scottish watershed to investigate how heterogeneous farming practices created landscape mosaics of varying ecosystem services. Although there are availability, consistency, and quality concerns, parcel data aligns well with the spatial extent of decision-making entities (e.g., individuals and households), and provides opportunities to analyze policy options at the level at which they would be enacted [72]. Use of parcel data for research and policy evaluation has grown across a wide range of fields [72], and represents a promising avenue in FEWS research.

Despite the consistent objective of better incorporating human behavior into FEWS research, only six studies (20%) leveraged survey data to inform their ABMs. Becu et al. [31] conducted a survey of farmers' crop choices in two villages in the same watershed in Thailand to simulate water use patterns and potential conflicts between upstream and downstream water users. Zhuge et al. [38] incorporated two large-scale surveys on travel behaviors and water-energy use to approximate individual water and energy consumption patterns in Beijing based on daily activities. In an application in the Mekong and

Niger River basins, Khan et al. [48] surveyed water users' priorities (e.g., hydropower, food production, ecosystem health), which were used to explore the impacts of different rankings on long-term water use patterns and related ecosystem services. Huang et al. [34] used a statewide survey of farmers' attitudes towards energy crop production to simulate how farmers' planting decisions translated into biofuel prices and environmental impacts. Guillem et al. [35] used phone interviews to administer a choice-based conjoint survey to estimate preferences for economic, environmental, and social attributes of alternative land use regimes, which were used to define farmer typologies and parameterize utility functions. Shastri et al. [43] used published survey results in the literature to also design a farmer typology, which distinguished between those that based their decisions only on economic data and those that additionally observed the actions of their peers. All but one of these ABM applications used survey results to represent individual entities (e.g., farmers, consumers). However, the models were applied to spatial extents ranging from small watersheds to a major metropolitan area to transnational river basins.

*3.2. Simulation Design*

3.2.1. Agents

Agents were characterized based on whether decision-making was active or reactive, and the spatial and temporal scales at which the entities represented as agents and their actions were implemented. The classification of active or reactive agents was based on the implementation of agents' decision-making rules, and followed the definition provided by Berglund [30]:

> "Reactive agents are defined as those that do not learn or update their rules of behavior but respond passively to other agents and the environment. ... They can be represented using a set of simple or complex rules to simulate their response to events in an environment. Alternatively, active agents are goal-directed and initiate actions to achieve individual goals. They may use an optimization methodology to select actions to satisfy a formalized goal or objective function." (p. 3)

The choice of active or reactive agent representation was hypothesized to vary with the spatial and temporal dimensions of the entities modeled. Specifically, active agent representations may be associated with ABMs at individual or household scales and with annual decision frequencies, since these roughly correspond with most empirical observations of human behavior and existing decision theories [73–76]. However, the choice of active or reactive agents was generally independent of the types of agents represented or the spatial disaggregation of the entities modeled. Most differences fell along lines of ABM versus MAS paradigms.

Overall, ABM applications were near evenly split between active (n = 15) and reactive (n = 14) agent representations. The proportion of applications that implemented reactive agents with time steps finer than monthly (43%) was much higher than that of applications with active agents with similar time steps (13%). This difference was in part due to the challenges of collecting data at sufficient temporal resolution to inform behavioral rules for such high frequency decisions and actions [77,78]. In contrast, the large majority of active agent implementations made decisions on annual time steps (87%), which reflected the large proportion (89%) of ABM applications representing some aspect of agricultural production and the yearly cadence of crop and/or land use decisions. All of the active agent implementations explicitly modeled farmer or related producer (e.g., biofuel refiner) decision-making. Ten of those applications were in food–energy or energy–food sectors concerning bioenergy crop production and markets. This is likely explained by the stated need to model agency (e.g., collective behavior) within supply networks and behavioral responses to dynamic prices and policy interventions associated with biofuel markets.

3.2.2. Agent and Model Environment Spatial and Temporal Scales

An overarching challenge for modeling interactions between social and ecological and/or natural resource systems is reconciling—both technically and conceptually—spatial and temporal scales of

system components, their interactions, and resulting coupled system dynamics [71,79]. This challenge is exacerbated in FEWS research because nexus issues (i.e., trade-offs) often emerge at larger spatial and longer temporal scales than the origins of production or consumption of FEW resources [1,4,16,80]. Thus, a defining feature of FEWS models is how the conceptualization of agents and their environments are matched with available data and key cross-sector interactions. Figure 3 shows the variety of approaches taken to aligning the spatial scales of agent representations and model environments and the time scales over which agents' behaviors and FEWS dynamics were modeled.

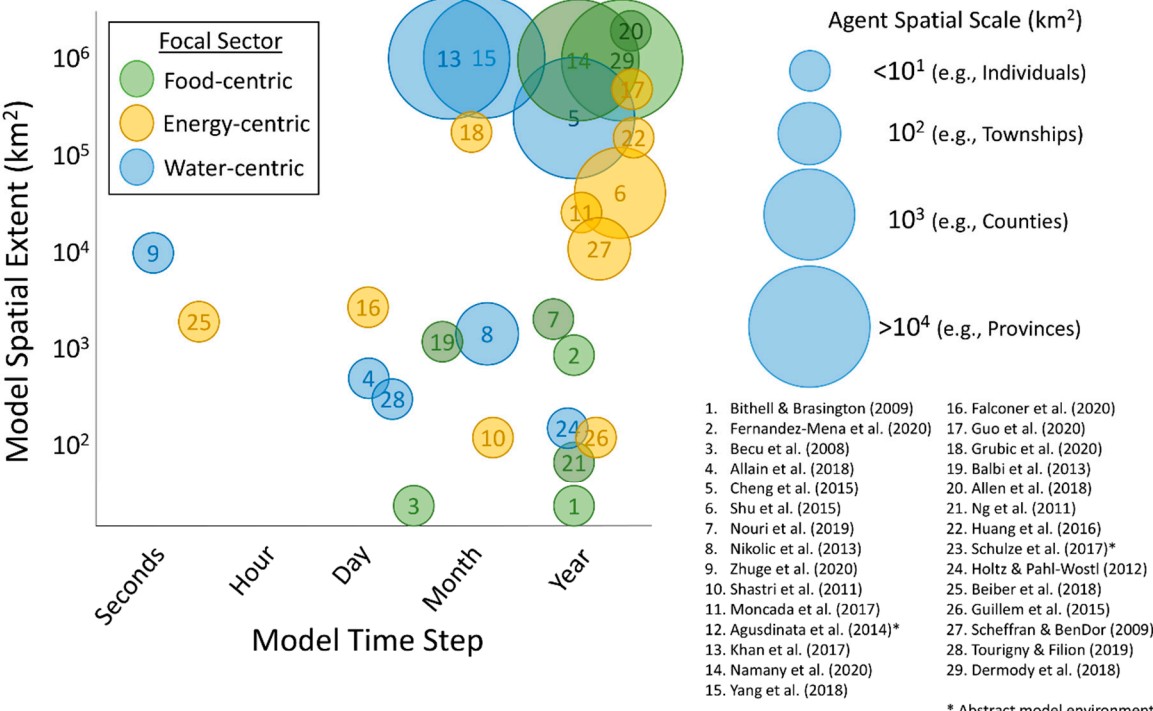

**Figure 3.** Configurations of model spatial extents and time steps and spatial scales of agents represented in agent-based model (ABM) application in food–energy–water system (FEWS) research. Circle size indicates the spatial scale of agent representations, and circle color is associated with the FEWS sector perspective from which the ABMs were designed. Notes: (1) when two or more models possessed the same spatial extent and time step, circles were offset to reduce occlusion. (2) ABM applications indicated with an asterisk implemented non-spatial, abstracted model environments, and thus were not plotted in the figure.

All but two of the ABM applications were spatially explicit, which was consistent with the desire to use ABMs to represent spatially disaggregated resource supply and/or demands. Agriculture-centric applications (i.e., food–energy or food–water) were skewed towards more disaggregated entities as agents (Figure 3), including individual farmers (n = 7) or households (n = 1). Two agriculture-centric applications implemented aggregate agents in the form of individual cities and their associated agricultural hinterlands [52] or individual agricultural importer or exporter countries [51]. Water-centric applications were more evenly split between disaggregated and aggregated agent representations, with three ABMs representing individuals or farms consuming water and four representing aggregate, multi-sector water users at river basin or administrative boundary scales. Energy-centric applications were also split with five applications focusing on individual electricity consumers or farmers producing bioenergy crops, and seven applications implementing a range of farmer aggregations and/or bioenergy production facilities.

No clear association was apparent between the focal FEWS sector and the spatial extent of the model application (Figure 3). The spatial extent of model environments ranged from small stream catchments of

roughly 4 km$^2$ to global trade networks. The largest spatial model extents (i.e., greater than 10,000 km$^2$) tended to be more recent applications occurring 2015 or later, whereas the smallest spatial extents (i.e., less than 100 km$^2$) were among the earliest applications occurring before 2012. This trend may indicate a recognition of the increasing scale of FEW system problems, and/or increasing computational power and attention to scaling challenges.

Similarly, there were no clear relationships between the stated, primary purpose of the ABMs and design choices of the ABMs spatial extents, spatial scales of the agents, and time steps (Figure 4). The majority of models focused on methodological aspects of ABM implementation chose smaller-scale agents (e.g., individuals or households) simulated at annual time steps. This reflects the dominance of agricultural applications. Models that were used to primarily improve system understanding were applied in relatively small spatial extents, with smaller-scale agents, and simulated at annual time steps. However, there was a wide range of variability in the model design choices based on models' stated purposes, which suggested that ABM applications in FEWS were to some degree context dependent.

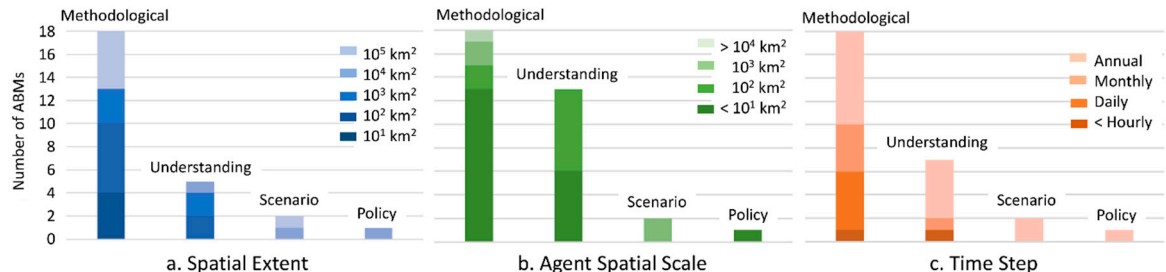

**Figure 4.** Relationships among the review models' primary purpose (methodological, system understanding, scenario assessment, and inform policy) and design choices for models' (**a**) spatial extent, (**b**) spatial scale of decision-making entities (i.e., agents), and (**c**) simulation time step.

Both ABMs that implemented abstracted or non-spatial model environments [42,57] did so for reasons of generality and broader applicability. In the case of Agusdinata et al. [42], agent interactions occurred within a non-spatial network of biofuel users, refineries, and farmers, and the conceptual focus was understanding the role of information uncertainty and delay in the exchange network in driving supply and demand dynamics. Schulze et al. [57] constructed a generalized model environment to be broadly applicable throughout various European contexts with the purpose of developing general mechanistic understanding of processes leading to or inhibiting the expansion of short rotation forestry.

The scale of agent representations was generally smaller than that of the spatial model extent, which reflected the need to simulate distributed activities or interactions and their supply and demand consequences. More aggregated agent representations were generally used with increased model application extent; however, there were also examples of both highly disparate and similar agent spatial scales and model extents. For example, Grubic et al. [37] investigated the effects of wider adoption of multiple micro-generation technologies by households across the entire United Kingdom to track resulting consumption of infrastructure resources, CO$_2$ emissions, and electricity costs. Allen et al. [33] explored the country-level sustainability implications of individual adoption of innovative food waste and energy efficiency technologies. On the other end of the spectrum, two applications at the provincial level in China represented agents aggregated to only one order of magnitude smaller than the spatial extent of the model. Shu et al. [46] modeled the potential for bioenergy crop expansion in Jiangsu Province with agents that represented 102 county-level administrative units aggregated into 70 counties. Cheng et al. [49] assessed food and water security scenarios under climate change for Heilongjiang Province with single sector agents that aggregated supply or demand for the entire province.

Overall, high temporal resolution and aggregated agent representations were generally traded-off with the richness of agent behavior that was represented. ABM applications with monthly time steps or finer tended to feature reactive agents used to generate realistic temporal patterns of supply or

demand. More sophisticated, active agent behaviors, such as risk aversion or social learning, were only present in ABMs with time steps of a month or longer. Moreover, ABM applications that relied on spatially aggregated agents to model larger spatial extents were either reactive or implemented with simplified, homogeneous behavioral models for all agents.

*3.3. Incorporating Human Behavior*

3.3.1. Agent Interactions

Many of the justifications for using ABMs invoked interactions among agents as key to understanding disaggregated supply or demand behaviors or the emergence of new system-level dynamics. However, direct interactions among agents were simulated in only 34% of ABM applications, while five (17%) ABMs include neither direct nor indirect agent interactions. Three of the ABMs that did not include either direct or indirect agent interactions used reactive agents modeled at high temporal resolutions. Zhuge et al. [38] modeled individuals and households as purely reactive agents conducting daily activities that generated spatially disaggregate and temporally explicit water and energy demands. Similarly, Beiber et al. [10] simulated both water and energy use patterns of individual consumers and companies to generate time-dependent demands. Allen et al. [33] used disaggregate data to approximate daily activities of households to generate annual food and energy demands, which were used to estimate residential sector GHG emission.

The remaining two ABMs that did not include agent interactions used active agents in agricultural contexts. Balbi et al. [45] simulated farmers' crop production and irrigation management decisions based on their own risk attitudes and endogenous price and climate forecasts. Guillem et al. [35] modeled crop rotation, spatial planting patterns, and management intensity of farmers of one of three attitudinal types. In both cases, the focus of the model was skewed towards the implications of macro-scale patterns that emerged from individual behaviors, such as GHG emissions from the agricultural sector [45] or wildlife habitat across the landscape [35], rather than the behaviors of individuals themselves. This provides a likely explanation as to why agent interactions were excluded despite having active agents and annual timescales for agent decisions.

Most ABM applications (69%) modeled indirect interactions among agents to represent spatially disaggregated production/consumption behaviors that have aggregate effects on a given resource system. Indirect agent interactions were competitive in nature, either through consumption of a common resource (e.g., surface water) or through supply and demand relationships in markets. For example, farmers' irrigation decisions affected water availability for downstream users [6,31] or for groundwater supply in future time steps [58]. Direct agent interactions were modeled either as regulator–regulated relationships (e.g., water withdrawal permits by regulatory agent), or as exchange interactions in supply and/or material flow networks. For example, both Nikolic et al. [47] and Nouri et al. [56] used a type of government agent to implement—reactively (i.e., prescribed regulatory action) and actively (i.e., selection among policy options based on an objective), respectively—water use restrictions and allocate water supply to various water users. Fernandez-Mena et al. [32] provided an example of combined direct and indirect agent interactions in which agents competed for materials based on price and scarcity, and engaged in production/consumption exchanges that created materials flows in a spatially constrained agro-food network.

3.3.2. Use of Behavioral Theories, Agent Motivations, and Types of Decisions

ABMs have been criticized for creating ad-hoc behavioral rules without grounding in existing behavioral theories, and when behavioral theories are used to inform agent decision-making, microeconomic theories of rational choice and profit maximization are most commonly used [73–75]. ABM applications in FEWS research showed the same tendencies. More than a third (n = 10) of ABM applications implemented agent actions and/or decisions without reference to or grounding in existing behavioral theories. Rational choice theory and profit maximizations were the most common

theoretical bases for agent decision-making in applications with active agents (11 of 15). The majority of these applications were in the food–energy or energy–food sectors. Four applications of active agents used alternative or additional decision-making or interaction theories. Tourigny and Filion [39] used social contagion theory to model the diffusion of efficient water use technology adoption in which agents' probabilities of adopting new technologies was influenced by connections in their social network. Guillem et al. [35] used aggregate nonlinear utility functions for each heterogeneous farmer type to prioritize economic, environmental, or social values in crop choice and land management decisions. Risk aversion was implemented in three applications [43,45,58] as a modification of rational choice theory using a utility maximization framework. Bithell and Brasington [36] used a satisficing decision model with which farming households in Nepal selected plots of land and made cropping decisions in order to meet household subsistence needs. Fernandez-Mena et al. [32] modeled producers and consumers within a local agro-food network using the assumption that actors would maximize the amount of materials or goods supplied or acquired and minimize waste accumulation to simulate material flows.

ABM applications that did not use behavioral theory to inform agent decisions relied on survey data or aggregate sector data to generate individual agent supply or demand behaviors that approximated observed trends. Aggregate data about household characteristics, farming region or watershed conditions, or city demographic were used to estimate distributions of agent behaviors or actions. For example, Khan et al. [48] and Yang et al. [8] assigned agents preference rankings derived from survey data for prioritizing water use between irrigation, hydropower generation, and ecological management uses, which were used as input into agents' decision-making in response to changing conditions. Agent actions were similarly prescribed in high spatial and/or temporal resolution models [6,10,37,38,55], to generate spatially disaggregated and high frequency demands for water and/or electricity.

### 3.3.3. Social Networks

Ten ABM applications explicitly modeled social networks, and were equally distributed across FEWS sectors. Social network representations showed a wide range of sophistication based on whether agents chose their social connections and the spatial constrains on possible connections. The simplest representations involved a social network spatially constrained to only immediate neighbors [34,47] among which information was exchanged and/or behavioral options were observed. Relaxing or expanding neighborhood assumptions produced a wide range of spatially structured social networks. For example, in Khan et al. [48] and Yang et al. [8], the river network spatially structured social networks, such that downstream water users could make direct requests of upstream water managers to increase water supply. Fernandez-Mena [32] simulated exchange networks linked by material flows. Agents chose their exchange partners within a spatial radius constrained by material transportation costs. Similarly, Dermody et al. [52] simulated infrastructure networks among cities and nearby agricultural hinterlands and their interactions with global trade networks connecting major cities to investigate food security and virtual water trade. Two applications used randomly generated small-world graphs to represent social network connections independent of agents' spatial relationships [37,39].

Social network influences also varied in how they informed agents' decisions. The most common application was to model imitation behavior among socially connected agents. For example, social network connections were modeled to simulate conformance behaviors among agents for decisions about whether to undertake an action, such as planting a specific crop [34,41,43] or adopting new water and energy efficient technologies [37]. Imitation behavior was modeled explicitly as an information or technology diffusion processes [81], such that a threshold of socially connected agents must have taken the action before a given agent also took the action. In bioenergy crop applications, for example, farmers checked how much area other farmers in their social networks had in bioenergy crop to inform their planning decisions. For value- or quantity-based decisions (e.g., price), agents weighted their own experiences relative to information available through peers within social networks when forming expectations for

future values [43]. Notably, none of the ABM applications used empirical data to implement social network structures or rules for network interactions (e.g., weighting personal vs. socially communicated information).

### 3.4. ABM Innovations

Several model design features unique to FEWS modeling and ABM practice were observed. In particular, multiple applications leveraged the bottom-up nature of ABMs to make the causal link between variations in agent-level information flow and social interactions to sector-level system dynamics. For example, Agusdinata et al. [42] and Balbi et al. [45] both explored how the accuracy and timing of information flows affected individual decisions with consequences for larger sectors. Agusdinata et al. [42] manipulated time lags in price information in biofuel supply chains to explore the implications of the differing planting decision that resulted. Balbi et al. [45] tested the effects of varying accuracy of climate forecasts on planting decisions and subsequent water use. Additionally, Holtz and Pahl-Wostl [58] implemented farmer-level risk aversion behavior and influences of path-dependence on decisions to adopt irrigation technology to reconstruct historical trends, and were able to distinguish between agency and structural drivers of historical groundwater use. In this case, the ABM was used for inverse modeling to demonstrate the incompleteness of behavioral explanations for water use patterns.

Similarly, the bottom-up nature of ABMs provided an opportunity to derive unique sector-scale sustainability metrics from explicitly modeling and aggregating agent-level interactions. Yang et al. [8] used the deficit concept to compare the impacts of agents' water use decisions and the trade-offs they presented across food, energy, and water sectors in large transboundary watersheds. Similarly, Beiber et al. [10] examined farmer-level opportunity costs of food production foregone to produce biofuels and/or divert water use to power generation to compare the overall impacts of alternative water supply and energy deployment policies. Schulze et al. [57] took a slightly different approach to exploring potential impacts of policy. In the context of short rotation forestry, the authors used a novel sensitivity analysis in which multiple landscapes with the same aggregate characteristics but different spatial configurations were implemented to test the effects of landscape structure on supply chain operations and production levels. This was a way of exploring the potential feasibility of supply chains across heterogeneous landscapes of the broader European geographic region. Due to the nascent development of short rotation forestry, Schulze et al. [57] modeled the adoption process explicitly to explore contextual influences on adoption decisions, rather than the more common approach of empirically or experimentally specifying adoption rates for a given practice or technology.

A distinct feature of some of the ABMs reviewed was the inclusion and explicit simulation of supply and production networks. Many ABM applications in agricultural or land-use change research domains address how consumer demands transform distant production landscapes, but supply chains and the structures they impose on agent interactions are rarely represented explicitly. However, supply chain modeling was prevalent in biofuel applications. Huang et al. [34] suggested that this focus within the energy–food sector was due to policy mandates to increase biofuel supply, and thus the logistics of bringing bioenergy crop supplies to market are salient. Only one ABM outside of bioenergy considered supply chains, Fernandez-Mena et al. [32], which explicitly modeled exchange relationships and material flows in agro-food networks. Supply chains were not a feature in the other models reviewed that considered agriculture (e.g., irrigation, farm water management) or energy (e.g., electricity generation).

Innovations in modeling the diversity of human behavior were also observed in several applications. ABM applications for socio-environmental systems commonly incorporate multiple dimensions of heterogeneity in agent attributes, but far fewer represent behavioral heterogeneity [73–75]. This was also true for FEWS ABM applications, as agent heterogeneity was typically applied as spatial heterogeneity. Two ABM applications were the exception and addressed behavioral heterogeneity by implementing heterogeneous agent types. Ng et al. [41] represented "cautious" and "bold" farmer types by manipulating the rate at which new information was incorporated into agent expectations

of future prices, costs, yields, and weather, which had direct influences on cropping decisions and subsequent surface water quality. Guillem et al. [35] modeled four different attitudinal types of farmers engaged in bioenergy crop production, which varied in their values for economic, environmental, and social objectives. Model experiments were conducted with all farmers of one type or represented in proportion to their abundance in survey data to infer the influence of each type on landscape-scale land-use patterns and wildlife habitat.

### 3.5. ABM Limitations

The most prominent limitation in ABM design stated by the articles' authors was oversimplified and/or prescribed agent behaviors. The absence of individual or social learning was frequently cited as a model limitation, which constrained the models' abilities to investigate potential adaptations to changing climate or policy conditions. Despite the recognition that other FEWS modeling approaches narrowly focus on economic rationales and/or use representative agents [4], non-economic influences on decision-making were consistently lacking, and aggregate agents (e.g., single water supply or energy use agents) were used to represent regional or sector-wide demands. When different agent types were represented, decision-making rules and/or behavioral options were assumed to be homogenous for all agents within a given type. Similarly, when microsimulation techniques were used to simulate highly disaggregated agent activities (i.e., individuals and time steps less than a month), resource consumption, or technology adoption decisions were purely reactive or even randomly made for all agents. In addition, all of the ABM applications classified as having reactive agents used prescribed behavioral rules by definition. While not necessarily a problem for models whose primary goals were to generate distributed supply/demand actions, this presented a significant limitation for scenario analysis. Many ABM applications prescribed or hard-coded agent responses to alternative policies or management regimes. For example, agents' values or preferences were often assumed to be static over time and did not consider social or system-level feedbacks that could change motivations in reality [35].

Despite a stated goal of representing disaggregate supply/demand behaviors, agent behaviors were not spatially resolved in many ABM applications. Rather, the spatial distribution of farmers and water sources was typically considered. When lacking, spatial relationships were deemed either not important for the model purpose, or abstracted in the form of spatially aggregated agents (e.g., one agent per river sub-basin, city, or province). In the former case, agents were assumed to be embedded within supply chains, and thus their position in the supply chain was a more relevant feature (e.g., [42,47,50]). In the latter case, efforts to scale-up distributed production or demands to entire regions or sectors drove the design of agents. In other situations, limitations associated with using aggregate data (e.g., household characteristics) to parameterize disaggregated agent consumption patterns resulted in non-spatial model designs.

The final recurrent limitation was insufficient data to parameterize agents' attributes and/or decision-making. Many ABM applications cited mismatches between readily available aggregate data and the individual-level characteristics needing specification in the models. For example, lacking sufficient data to assign specific farmer types to individual parcels, Guillem et al. [35] randomly allocated farmer types to farm parcels in proportion to their presence in the population, and used multiple model replications to account for stochasticity in landscape-scale outcomes. Similarly, Allen et al. [33] had to make many correlative rather than causal assumptions about relationships between household characteristics and consumption patterns. Parameterizing agents' decision-making was also a stated limitation for applications that endeavored to simulate active agents and their behaviors. This resulted in limited consideration of non-economic decision factors, such as risk aversion, collective behaviors, and subjective norms. Indeed, the inherent difficulties in collecting and implementing data about decision-making processes has been highlighted as one of the eight grand challenges for socio-environmental systems modeling [71].

## 4. Discussion

Overall, the innovative ABM elements reviewed represent progress towards integrating human behavior and other social dimensions into ABM practice and FEWS research more broadly. However, the FEWS lens also led to modeling design choices that limited the potential value-added of the distributed, bottom-up decision-making simulated with the agent-based approach. One such limitation was the imposition of the FEWS framework on agents' decision-making and which sources of information were deemed relevant to agents' decisions. For example, in agricultural-focused applications, farmers' decisions often considered resource use efficiency and/or sustainability, but real-world farming decisions typically do not consider FEWS interconnections in these terms when choosing crops or irrigation rates. In many cases, such considerations were incorporated into the behavioral rules of reactive agents in order to explore potential FEWS trajectories under alternative scenarios. This approach was prevalent in water-energy applications where agents' decisions to adopt more efficient or sustainable technologies were purely reactive. A way to incorporate human decision-making more realistically into FEWS analysis is to adhere to the bounded rationality paradigm. Agents' choices among behavioral alternatives should be based on individual motivations, valuations, and information, rather than imposing FEWS-level valuations of behavioral alternatives. The emergent consequences of agents' actions and their feedbacks with larger-scale FEWS dynamics can then be simulated without 'baking in' cross-scale interactions.

The FEWS framework also challenges agent representations that are consistent with the spatial and temporal scales of disaggregated production and consumption behaviors, system-level change in FEW sectors, and feedbacks in between [4,71,79]. Figure 3 shows the range of spatial aggregations used to represent agents, which varied from individuals to entire countries. In some cases, the choice of modeling unit was environmentally rather than socially informed (e.g., watersheds versus irrigation districts). While this may be practical for tracking resource flows and budgets and/or integrating ABMs with other models, such design choices may not reflect the spatial or political dimensions of actual decision-making and interactions [16,79]. People typically interact through formal and informal institutions (e.g., regulations, social norms) and social gathering locations (e.g., farmer cooperatives, workplaces) that often do not conform to the structure of natural systems. In addition, interaction structures, such as social networks or daily activities leading to social encounters, are difficult to observe, and agent attribute data is often aggregated to decouple individual characteristics from spatial location. Nearly half of the ABM applications with agents at the scale of individual, household, or farm(er)s dealt with limited data or knowledge of decision-making process by using reactive agents or aggregate agent representations. While these may be practical design choices, they do not advance the representation of actual decision-making or human behavior in the FEWS context.

Such design choices may also be driven by the demands of ABM integration with other models (often of different paradigms) to represent cross-sector influences and/or cover large geographic extents with agents that aggregate many individual entities. Aggregation often limits the behavioral richness that can be simulated in an ABM. Such coupling challenges between human and natural systems models are well documented and discussed elsewhere [71,79,82]. In the FEWS context, however, there is relatively little critical discussion or reflection about best practices for scaling-up individual behaviors to aggregate levels. Most scaling decisions in the applications reviewed appeared to be technically driven (i.e., supporting model integration) rather than conceptually driven. Moreover, human systems components (e.g., attributional or behavioral heterogeneity) were abstracted or aggregated more often than natural system components (e.g., spatial environmental heterogeneity). More innovative approaches to model upscaling or downscaling are needed to maintain behavioral richness and realism while meeting the technical challenges of large-scale and cross-sector analyses [71].

Finally, modeling two or more FEWS and their interconnections poses fundamentally new model evaluation and validation challenges. A common criticism of ABMs is that they are data intensive, because modeling system dynamics at the individual level entails many free parameters, and decision models and agent attributes must be calibrated [83]. The FEWS modeling challenge presents a similar but reverse challenge [80]—the diversity of actors across different sectors, and often large spatial extents

of application increase data demands. Moreover, the strong sustainability focus of FEWS research means ABMs are often used in scenario analysis to explore potential shifts in production/consumption patterns in response to policy interventions or climate change impacts. However, as Allen et al. [33] point out: "modelling and exploring 'possible futures' for a 'real' situation, such as the material and energy flows of households, requires at the very least the inclusion of variables and interactions concerning the ecosystem, environment, economy, local culture, climate and technological changes" (p. 342). In other words, not only is behavioral information needed to simulation decision-making, FEWS modeling requires such data for diverse agents across multiple sectors and/or larger geographic extents. Several of the reviewed ABM applications discussed limitations due to infeasibility of gathering agent-level for these reasons, and how this led to simplified representations of agents' behaviors [32,39,45]. Relatedly, large-scale, multi-sector models also pose substantial challenges for sensitivity analysis and model evaluation. Bithell and Brasington [36] explain that the increase in model components necessary to represent cross-sector interactions concurrently expands the parameter space that must be explored via global sensitivity analysis, and non-linearities may manifest in coupled model outcomes that are not expressed by each model individually. Model outputs can reach the complexity and scale of "big data" requiring specialized tools to analyze spatially explicit and/or combinatorially complex parameter outcomes [45,71].

## 5. Conclusions

The need to integrate richer and more realistic human behavior into FEWS research has been recognized, and ABMs are increasingly filling that gap. The growing use of ABMs has been driven in large part by the importance of representing decentralized, spatially heterogeneous, and time-varying production and consumptions behaviors in FEWS interactions. Disaggregation of sector-level supply and demand for FEWS resources is particularly important for addressing questions about the potential impacts of regulation or policy interventions and changing climate, and how responses to those changes might vary over space and time. Secondarily, there is a recognition of the importance of human behavior in FEWS dynamics for producing non-linear and potentially sub-optimal outcomes. However, progress towards incorporating the "social" was mixed with more than a third of models lacking a grounding in behavioral theory, which mimics broader trends in ABM applications in socio-environmental systems research [73–75]. Similarly, only a third of ABM applications reviewed included direct interactions among agents. Learning from the complex landscape of ABM applications in FEWS research, this review identifies three specific future research directions.

Explicitly modeling supply chains or networks brings greater resolution to the structure of agent interactions, and how market- or sector-level trends can emerge from distributed decision-making. This represents a current research gap in ABMs for socio-ecological systems modeling, and an area where ABM in FEWS research is more advanced. Although the inclusion and explicit simulation of supply and production networks was prevalent in ABM FEWS applications, this remains an area that could be enriched by broader ABM practice. There is a robust history of ABM, or multi-agent systems (MAS), applications to model logistics networks and commodity supply chains as complex adaptive systems [84–86], yet there has been little cross-pollination with FEWS research. This body of knowledge and ABM practice could be productively integrated into FEWS research, and socio-ecological systems modeling more broadly, to better model the structures directly responsible for connecting producers and consumers.

Social science research methods were also generally lacking in support of ABM design and implementation. Certainly, the scale of some of the ABM applications imposed constraints on the types and depth of data that could be collected about decision-making and behavioral motivations. Such data is more feasible to obtain in smaller-scale investigations, yet few ABM applications pursued this avenue despite noting the lack of behavioral or cognitive information as an ABM limitation. In particular, power differentials among FEWS stakeholders are recognized as a strong influence on cross-sector interactions and the feasibility of policy interventions, for which qualitative and contextually rich

research methods are needed to generate information that can be translated into ABMs [71]. Calls were made within the reviewed articles to use methods such as interviews and/or role-playing games to obtain empirical data about individual motivations and decision-making processes [39,41]. However, such approaches are not yet common in FEWS research, and future efforts would benefit from leveraging social science methodologies to inform ABM design and implementation.

Finally, ABMs were often used to make the connections between sectors by translating the demands for one resource (e.g., water) into the consumption of another (e.g., energy) through agents' actions. The agent-based approach is grounded in fundamentals of complexity science to model complex systems, and thus the structure of interactions encoded in ABMs is reflective of that of the real systems being modeled. The structure of the ABMs could be used productively to identify important processes and/or structural relationships that are present in the real FEWS and may be important to include in any FEWS research (i.e., not just modeling). Moreover, comparing the structure of ABMs applied to the same FEWS has the potential to generate insights into common and unique features and functioning of those systems. Such an approach has been used successfully with land-use change ABMs [87] and is promising in the FEWS research domain.

**Supplementary Materials:** The following are available online at http://www.mdpi.com/2073-445X/9/12/519/s1, Supplementary Table S1: Article coding.

**Funding:** This research was funded by the National Science Foundation, INFEWS 1856054.

**Conflicts of Interest:** The author declares no conflict of interest.

## Appendix A

Target set of articles to ensure a comprehensive search of the literature.

1. Bell, A.R.; Ward, P.S.; Shah, M.A.A. Increased water charges improve efficiency and equity in an irrigation system. Ecol. Soc. 2016, 21, doi:10.5751/ES-08642-210323.
2. Bieber, N.; Ker, J.H.; Wang, X.; Triantafyllidis, C.; van Dam, K.H.; Koppelaar, R.H.E.M.; Shah, N. Sustainable planning of the energy-water-food nexus using decision making tools. Energy Policy 2018, 113, 584–607, doi:10.1016/j.enpol.2017.11.037.
3. Farhadi, S.; Nikoo, M.R.; Rakhshandehroo, G.R.; Akhbari, M.; Alizadeh, M.R. An agent-based-nash modeling framework for sustainable groundwater management: A case study. Agric. Water Manag. 2016, 177, 348–358, doi:10.1016/j.agwat.2016.08.018.
4. Fernandez-Mena, H.; Gaudou, B.; Pellerin, S.; MacDonald, G.K.; Nesme, T. Flows in Agro-food Networks (FAN): An agent-based model to simulate local agricultural material flows. Agric. Syst. 2020, 180, 102718, doi:https://doi.org/10.1016/j.agsy.2019.102718.
5. Holtz, G.; Pahl-Wostl, C. An agent-based model of groundwater over-exploitation in the Upper Guadiana, Spain. Reg. Environ. Chang. 2012, 12, 95–121, doi:10.1007/s10113-011-0238-5.
6. Khan, H.F.; Yang, Y.C.E.; Xie, H.; Ringler, C. A coupled modeling framework for sustainable watershed management in transboundary river basins. Hydrol. EARTH Syst. Sci. 2017, 21, 6275–6288, doi:10.5194/hess-21-6275-2017.

7. Mo, W.; Lu, Z.; Dilkina, B.; Gardner, K.H.; Huang, J.-C.; Foreman, M.C. Sustainable and Resilient Design of Interdependent Water and Energy Systems: A Conceptual Modeling Framework for Tackling Complexities at the Infrastructure-Human-Resource Nexus. SUSTAINABILITY 2018, 10, doi:10.3390/su10061845.

8. Namany, S.; Govindan, R.; Alfagih, L.; McKay, G.; Al-Ansari, T. Sustainable food security decision-making: An agent-based modelling approach. J. Clean. Prod. 2020, 255, doi:10.1016/j.jclepro.2020.120296.

9. Ng, T.L.; Eheart, J.W.; Cai, X.; Braden, J.B. An agent-based model of farmer decision-making and water quality impacts at the watershed scale under markets for carbon allowances and a second-generation biofuel crop. Water Resour. Res. 2011, 47, doi:10.1029/2011WR010399.

10. Perello-Moragues, A.; Noriega, P.; Poch, M. Modelling contingent technology adoption in farming irrigation communities. JASSS 2019, 22, doi:10.18564/jasss.4100.

11. Utomo, D.S.; Onggo, B.S.; Eldridge, S. Applications of agent-based modelling and simulation in the agri-food supply chains. Eur. J. Oper. Res. 2018, 269, 794–805.

## Appendix B

Publication outlets of reviewed agent-based modeling (ABM) articles.

The number of ABM applications in FEWS research has been increasing over the last decade, and nearly half of the reviewed articles were published in the last three years (Figure A1). Roughly 20% of articles were published in 2020. The earliest publication was in 2008 [31], which predated the Bonn Nexus Conference 2011 that approximates the point in time when the FEW nexus became a mainstream research and policy topic. The Journal of Cleaner Production was the most frequent outlet and included both food- and energy-oriented applications. Agricultural Systems, Energy, and Water Resources Management each had two publications apiece, and the remaining journals had only a single publication.

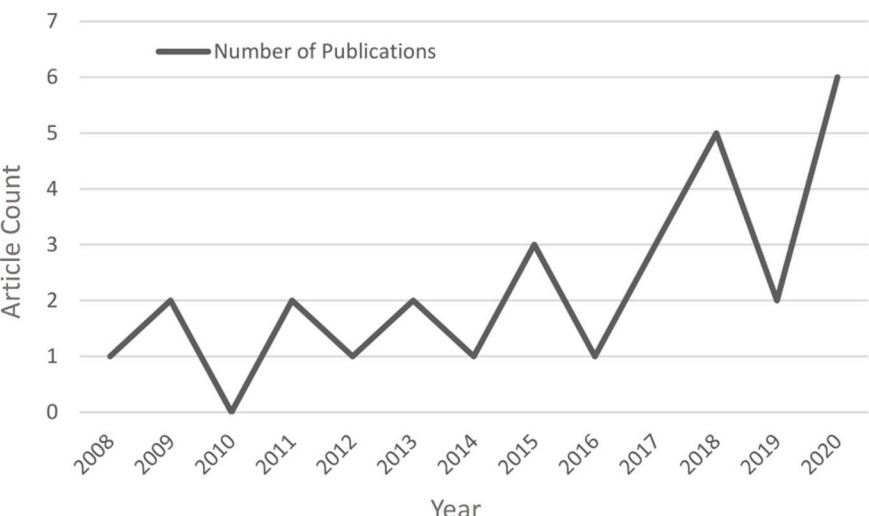

**Figure A1.** Publications of agent-based modeling (ABM) applications in the food–energy–water systems (FEWS) research domain per year (2020 includes search results up through June of that year).

**Table A1.** Journals in which reviewed agent-based models were published.

| Journal | Count | Journal | Count |
|---|---|---|---|
| Journal of Cleaner Production | 3 | Global Change Biology—Bioenergy | 1 |
| Agricultural Systems | 2 | Hydrology and Earth System Sciences | 1 |
| Energy | 2 | Int. J. of Environment and Pollution | 1 |
| Water Resources Management | 2 | Italian Journal of Agronomy | 1 |
| Agronomy for Sustainable Development | 1 | Journal of Water Resources Planning and Management | 1 |
| Applied Energy | 1 | Land Use Policy | 1 |
| Bioenergy Research | 1 | Natural Hazards | 1 |
| Earth's Future | 1 | Science of the Total Environment | 1 |
| Earth System Dynamics | 1 | Simulation | 1 |
| Energy Policy | 1 | Sustainability Science | 1 |
| Energies | 1 | Regional Environmental Change | 1 |
| Environmental Modeling and Software | 1 | Water Resources Research | 1 |
| | **Total** | | **29** |

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
