# Peer review of "Agent-Based Modeling for Integrating Human Behavior into the Food–Energy–Water Nexus"

_land, doi:10.3390/land9120519_

Round 1
Reviewer 1 Report
This review paper deals with an interesting topic and the review itself seems to be carried out in a structural manner. And although there is – in my view – nothing that speaks against this paper, there is also nothing that speaks in favour of it. In spite of the fact that it is well-written (apart from a double word in line 97 and absent use of super- and subscripts in km2 and CO2) it is still a hard read. The paper is long and always remains very descriptive. I consider myself reasonably representative for the audience that is targeted, and for me it is hard to stay focused. The review is scrutinous, but it is unclear to me why it is interesting to read who used which arguments, modelled which agents, used which timesteps, etc., at least not in the detail and length at which this is currently done. The discussion simply continues at the same descriptive level as the results section, and only in the conclusion a few more analytical observations are done. Perhaps there are more such analytical statements in the discussion, but then they got buried in the large amount of factual observations. All this makes the "so what?" question pop up constantly while reading the paper.
My advice would therefore be to reduce the length of the descriptive parts (use more tables and figures; those that are present are nice and informative!), and spend more words on an actual analysis of the findings. E.g. a cluster analysis or a correlational analysis, to see if certain modelling choices coincide with each other or with the modelling purpose or subject. Such an analysis helps to reflect upon the state-of-the-art, and can help to identify white spots in ABM for FEWS.
Author Response
Reviewer #1
Comment 1: This review paper deals with an interesting topic and the review itself seems to be carried out in a structural manner. And although there is – in my view – nothing that speaks against this paper, there is also nothing that speaks in favour of it. In spite of the fact that it is well-written (apart from a double word in line 97 and absent use of super- and subscripts in km2 and CO2) it is still a hard read. The paper is long and always remains very descriptive.
Response: Thank you for your constructive comments. The typos identified have been corrected. With regards to the length and descriptive nature of the manuscript, redundancies in descriptive statements have been removed and substantial revisions have been made to the Discussion and Conclusion sections. The Discussion and Conclusion sections have been restructured so that the former contains less descriptive elements and more synthesis, while the latter is focused on drawing conclusions from the reviewed work to inform future research directions in ABM applications in FEWS and socio-ecological system modeling more generally. After these revisions, the main text of the manuscript stands at 19.5 pages, which is a half-page shorter than the minimum length for review articles specified by Land.
Comment 2: The review is scrutinous, but it is unclear to me why it is interesting to read, who used which arguments, modelled which agents, used which timesteps, etc., at least not in the detail and length at which this is currently done. The discussion simply continues at the same descriptive level as the results section, and only in the conclusion a few more analytical observations are done. Perhaps there are more such analytical statements in the discussion, but then they got buried in the large amount of factual observations. All this makes the "so what?" question pop up constantly while reading the paper.
Response: To bolster the analytical elements in the manuscript, a new figure 4 has been added that enable comparison of model design choices (spatial extent, spatial scale of the agents, and time step) with the primary, stated purposes of the model. The associated text was also added at lines 491-499:
“Similarly, there were no clear relationships between the stated, primary purpose of the ABMs and design choices of the ABMs spatial extents, spatial scales of the agents, and time steps (Fig. 4). The majority of models focused on methodological aspects of ABM implementation chose smaller-scale agents (e.g., individuals or households) simulated at annual time steps. This reflects the dominance of agricultural applications. Models that were used to primarily improve system understanding were applied in relatively small spatial extents, with smaller-scale agents, and simulated at annual time steps. However, there was a wide range of variability in the model design choices based on models’ stated purposes, which suggested that ABM applications in FEWS were to some degree context dependent.”
With regards to the questions of “who ... modelled which agents, used which timesteps”, I would like to direct the reviewer’s attention to figure 3, which graphically displays relationships between spatial scales of agents and time steps simulated and links each item in the figure to its respective article. In addition, the text in section 3.2.2. links to and elaborates on the figure by providing the specific agents modeled associated with the scales depicted in figure 3.
With respect to “who used which arguments”, section 3.1.1. speaks directly to the justifications reviewed articles’ authors used in choosing the agent-based modeling (ABM) approach. Specific examples and associated references are provided in the text of that section. Section 3.1.2. and Table 2 further elaborate on the arguments for using ABM and the advances the approach offered with regards to model purpose. Table 2 lists references to the specific articles, and the full coding for all articles is provided as supplementary material. The authors’ arguments related to the implications of food-energy-water systems (FEWS) interactions and/or specific policies or scenarios were not considered in the scope of this review. The focus here was on the conceptualization, use, and implementation of ABMs in FEWS research. The following statement was added in lines 93-95 to clarify this scope:
“This review represents a horizon-scanning effort to consolidation information about the design, implementation, and motivations for using ABMs in FEWS research, and does not address the broader implications of specific FEWS interactions, policies, and/or scenarios.”
Comment 3: My advice would therefore be to reduce the length of the descriptive parts (use more tables and figures; those that are present are nice and informative!), and spend more words on an actual analysis of the findings. E.g. a cluster analysis or a correlational analysis, to see if certain modelling choices coincide with each other or with the modelling purpose or subject. Such an analysis helps to reflect upon the state-of-the-art, and can help to identify white spots in ABM for FEWS.
Response: Thank you for these constructive comments. Based on these and suggestions of other reviewers, the Discussion and Conclusion sections have been restructured so that the former contains less descriptive elements and more synthesis, while the latter is focused on drawing conclusions from the reviewed work to inform future research directions in ABM applications in FEWS and socio-ecological system modeling more generally. Additionally, a new figure 4 has been added that compares each models’ stated purposes with their design choices of spatial extent, agents’ spatial scale, and time step. There are no clear patterns in model purpose and design choices, which is interpreted to indicate context dependency in model design (mostly) independent of purpose.
With regards to the length, the main text now stands at 19.5 pages, which is a half-page shorter than the minimum page limit specified for review articles in Land.
Reviewer 2 Report
Tha manuscript "Agent-Based Modeling for Integrating Human Behavior into the Food-Energy-Water Nexus" is a review of literature concerning agent decision-making and behavior inclusion in the food/energy/water nexus.
While the manuscript topic is of high potential interest, I find it quite verbose and limited in its conclusions. IN particular, I find that Section 2 is not that necessary: it describes in excessive detail what should be a normal process of literature search and selection. This section could be summarized and joined with subsection 3.1 (Figures 2 and Table 1 are not essential).
Section 3.1.2 describes the purpose of the models described in literature. It is hard to follow and does not explain clearly the differences between various models proposed. Perhaps a summarizing table with model characteristics would be more helpful in that sense.
Section 3.2.1 could be more specific in describing the way in which agents' behavior is included in the models (structure of models).
The discussion section should attempt to look into models' outcomes and compare them critically. If a reader would examine this paper to get hints about the choice of a suitable model in a specific situation, which would in fact be a desirable conclusion, in my opinion, he would not get much valuable information.
The conclusions seem a repetition of concepts previously expressed elsewhere, Conclusions should be clear and concise, express findings of the study and they usually do not contain references. Some of the material in conclusions should be moved to the discussion section.
The author should attempt to give the manuscript a more critical cut and include material that could help readers make informed decisions on the possible selection of a suitable model.
Author Response
Reviewer #2
Comment 1: While the manuscript topic is of high potential interest, I find it quite verbose and limited in its conclusions.
Response: The manuscript has been trimmed, particularly in the Discussion and Conclusion sections, to make the arguments more concise. In the former section, revisions have been made to draw more synthetic points from the trends evident in the preceding sections. The Conclusions section has been tightened to focus on the three recommendations for future research in ABM applications of FEWS research: 1) advancing supply chain research in FEWS with ABMs; 2) increased use of social science research methods to inform model design and parameterization; and 3) comparative analysis of ABM structures to provide insight into structures of food, energy, and water systems and their component interactions and feedbacks.
Comment 2: IN particular, I find that Section 2 is not that necessary: it describes in excessive detail what should be a normal process of literature search and selection. This section could be summarized and joined with subsection 3.1 (Figures 2 and Table 1 are not essential).
Response: Thank you for the suggestion, but I respectfully disagree. The content of section 2 conforms with the reporting standards for conducting systematic reviews as described by the PRISMA statement (Moher et al., 2015) and related guidance from the socio-ecological systems research field (Magliocca et al., 2015). In addition, figure 2 and table 1 provide analyses of the research context in which the ABM applications were used, which answers one of the review’s stated research questions. However, for the sake of clarity and shortening the manuscript, figure 2 and table 1 have been moved to Appendix 2.
Magliocca, N. R., Rudel, T. K., Verburg, P. H., McConnell, W. J., Mertz, O., Gerstner, K., ... & Ellis, E. C. (2015). Synthesis in land change science: methodological patterns, challenges, and guidelines. Regional environmental change, 15(2), 211-226.
Moher, D., Shamseer, L., Clarke, M., Ghersi, D., Liberati, A., Petticrew, M., ... & Stewart, L. A. (2015). Preferred reporting items for systematic review and meta-analysis protocols (PRISMA-P) 2015 statement. Systematic reviews, 4(1), 1.
Comment 3: Section 3.1.2 describes the purpose of the models described in literature. It is hard to follow and does not explain clearly the differences between various models proposed. Perhaps a summarizing table with model characteristics would be more helpful in that sense.
Response: A new figure 4 has been added that compares specific model design choices of spatial extent, agents’ spatial scale, and time step with the primary model purpose stated by the authors. As the figure shows, there are no clear relationships between the purpose of model development and the structure of the ABMs. This interpreted to indicate context dependency in model design (mostly) independent of purpose. Given this finding, it is not a straightforward exercise to characterize, group, or otherwise classify the reviewed models by their stated purpose alone, which limits the usefulness of a summary table in this section.
Comment 4: Section 3.2.1 could be more specific in describing the way in which agents' behavior is included in the models (structure of models).
Response: Section 3.2.1 introduces the distinction between ‘active’ and ‘reactive’ agent representations. Specific representations of agent behavior vary substantially within these broad categories. For example, in the ways agents interact with other agents and their environments, individual motivations, and the heuristic or optimization models used to choose among behavioral alternatives. Each of these aspects of agent behavior are addressed in detail in section 3.3.
Comment 5: The discussion section should attempt to look into models' outcomes and compare them critically. If a reader would examine this paper to get hints about the choice of a suitable model in a specific situation, which would in fact be a desirable conclusion, in my opinion, he would not get much valuable information.
Response: ‘Model outcomes’ is a multi-layered and ambiguous term that can be interpreted to mean accuracy of model outputs compared with empirical data, specific spatial or temporal patterns of agent behaviors, and/or implications of model findings. Indeed, not all of the reviewed models strive to address all of these possible interpretations of ‘model outcomes’, and thus it is not straightforward (nor desirable) to compare all models based on a particular model outcome.
More importantly, the scope of this review is to describe the specific research contexts and aspects of FEWS that ABM have been used to study, and synthesize patterns and insights about the associated model design choices. Therefore, analyzing the outputs, performance, and implications of the models is beyond the scope of this review. Lines 91-95 have been revised to state clearly the intended scope of the review:
“Building on needs identified by recent reviews of FEWS modeling and related ABM approaches, this review considers both the research contexts and simulation designs of ABM applications in FEWS research. This review represents a horizon-scanning effort to consolidation information about the design, implementation, and motivations for using ABMs in FEWS research, and does not address the broader implications of specific FEWS interactions, policies, and/or scenarios.”
Comment 6: The conclusions seem a repetition of concepts previously expressed elsewhere, Conclusions should be clear and concise, express findings of the study and they usually do not contain references. Some of the material in conclusions should be moved to the discussion section.
Response: Thank you for this constructive suggestion. The Discussion and Conclusion sections have been revised exactly as suggested. Redundancies with previous text have been removed, and the Conclusions focus on future research directions. The references that persist in the Conclusions connect to the broader socio-ecological systems modeling literature, and thus establish the similarities and differences with the FEWS ABM literature, and have been cited previously in the text before this section.
Comment 7: The author should attempt to give the manuscript a more critical cut and include material that could help readers make informed decisions on the possible selection of a suitable model.
Response: The manuscript has been shortened by removing redundant sections, and revision have been made to the Discussion to make more synthetic points to inform future modeling efforts. For example, lines 740-745 have been added to help navigate the choice of ‘active’ versus ‘reactive’ agent representations:
“A way to incorporate human decision-making more realistically into FEWS analysis is to adhere to the bounded rationality paradigm. Agents’ choices among behavioral alternatives should be based on individual motivations, valuations, and information, rather than imposing FEWS-level valuations of behavioral alternatives. The emergent consequences of agents’ actions and their feedbacks with larger-scale FEWS dynamics can then be simulated without ‘baking in’ cross-scale interactions.”
Reviewer 3 Report
This is a very solid review of a focused area of literature. In that sense it is very useful as a resource to researchers.
As a social scientist working in this space I am very happy to see the limitations of existing models in including social science data. It might be worth emphasising this even more explicitly and more so the difficulties of modelling the complexity of power differentials between stakeholders. Perhaps modelling these is actually almost impossible and we need to be realistic about that and rather engage in iterative dialogue rather than hope than magnificently complex models can provide 'rational answers'.
Author Response
Comment 1: This is a very solid review of a focused area of literature. In that sense it is very useful as a resource to researchers.
Response: Thank you for your supportive comment.
Comment 2: As a social scientist working in this space I am very happy to see the limitations of existing models in including social science data. It might be worth emphasising this even more explicitly and more so the difficulties of modelling the complexity of power differentials between stakeholders. Perhaps modelling these is actually almost impossible and we need to be realistic about that and rather engage in iterative dialogue rather than hope than magnificently complex models can provide 'rational answers'.
Response: Agreed. Lines 829-832 have been added to emphasize this point: “In particular, power differentials among FEWS stakeholders are recognized as a strong influence on cross-sector interactions and the feasibility of policy interventions, for which qualitative and contextually rich research methods are needed to generate information that can be translated into an ABMs.”
Reviewer 4 Report
The article deals with crucial issue of the systematic review identified 29 articles in which at least two food, energy, or water sectors were explicitly considered with an ABM and/or ABM-coupled modelling approach. Agent decision-making and behaviour ranging from reactive to active, motivated by primarily economic objectives to multi-criteria in nature, and implemented with individual-based to highly aggregated entities. However, a significant proportion of models did not contain agent interactions, or did not base agent decision-making on existing behavioural theories. Model design choices imposed by data limitations, structural requirements for coupling with other simulation models, or spatial and/or temporal scales of application resulted in agent representations lacking explicit decision-making processes or social interactions. In contrast, several methodological innovations were also noted, which were catalysed by the challenges associated with developing multi-scale, cross-sector models. The introduction of ABM into the FEWS research domain has injected a more process-based perspective on social dimensions in FEWS research, yet many opportunities for more behaviourally rich agent-based modelling in the FEWS context remain. A review is prepared in good manner and consists of four figures, which are prepared by the author of the manuscript. Describe the axis of the Figure 2. Please change the presented graph into the column one. Line 192. Figure 2. Publications of agent-based modeling (ABM) applications in the food-energy-water systems 193 (FEWS) research domain per year (2020 includes search results up through June of that year).
Author Response
Comment 1: Describe the axis of the Figure 2. Please change the presented graph into the column one. Line 192. Figure 2. Publications of agent-based modeling (ABM) applications in the food-energy-water systems 193 (FEWS) research domain per year (2020 includes search results up through June of that year).
Response: Axes labels have been added to Figure 2. Figure 2 and Table 1 have been moved to Appendix 2. It is unclear what additional corrections the reviewer is requesting.
Round 2
Reviewer 1 Report
The author has improved the paper based on the reviews. Not as extensive as I would have liked to see (and I would personally have presented the new figure in a cross-table as this is a more common way to display the joint distribution of two nominal variables), and I do think the paper could have gained more in terms of 'take home messages' if the author would have taken the improvements a bit further. Nevertheless, the paper is well written and the review has been thorough. I have therefore no objections to it being published.
Reviewer 2 Report
The revised version of the manuscript and the author's comments addressed my previous queries.
The revised manuscript can be accepted for publication